# NURBS Interpolator with Minimum Feedrate Fluctuation Based on Two-Level Parameter Compensation

**DOI:** 10.3390/s23083789

**Published:** 2023-04-07

**Authors:** Mingxing Nie, Tao Zhu, Yue Li

**Affiliations:** School of Computer Science, University of South China, Hengyang 421001, China; niemx@usc.edu.cn (M.N.); tzhu@usc.edu.cn (T.Z.)

**Keywords:** NURBS interpolator, feedrate fluctuation, parameter compensation, curvature sensitive area

## Abstract

Feedrate plays a crucial role in determining the machining quality, tool life, and machining time. Thus, this research aimed to improve the accuracy of NURBS interpolator systems by minimizing feedrate fluctuations during CNC machining. Previous studies have proposed various methods to minimize these fluctuations. However, these methods often require complex calculations and are not suitable for real-time and high-precision machining applications. Given the sensitivity of the curvature-sensitive region to feedrate variations, this paper proposed a two-level parameter compensation method to eliminate the feedrate fluctuation. First, in order to address federate fluctuations in non-curvature sensitive areas with low computational costs, we employed the first-level parameter compensation (FLPC) using the Taylor series expansion method. This compensation allows us to achieve a chord trajectory for the new interpolation point that matches the original arc trajectory. Second, even in curvature-sensitive areas, feedrate fluctuations can still occur because of truncation errors in the first-level parameter compensation. To address this, we employed the Secant-based method for second-level parameter compensation (SLPC), which does not require derivative calculations and can regulate feedrate fluctuation within the fluctuation tolerance. Finally, we applied the proposed method to the simulation of butterfly-shaped NURBS curves. These simulations demonstrated that our method achieved maximum feedrate fluctuation rates below 0.01% with an average computational time of 360 us, which is sufficient for high-precision and real-time machining. Additionally, our method outperformed four other feedrate fluctuation elimination methods, highlighting its feasibility and effectiveness.

## 1. Introduction

In computer-aided manufacturing, Non-Uniform Rational B-Spline (NURBS) interpolation is used to generate a toolpath for the CNC machining of parts. Feedrate is an important parameter that needs to be carefully controlled during machining, as it affects not only the quality and accuracy of the machined part but also the tool life and the time required for machining. The feedrate of CNC machines is determined based on the NURBS curve and the cutting parameters such as spindle speed, depth of cut, and feed per tooth. However, the feedrate can fluctuate because of irregularities in the NURBS curve, which can result in unwanted effects such as tool vibration, chatter, and excessive tool wear. Therefore, minimizing feedrate fluctuations in the NURBS interpolation process is essential to attain high-precision and high-quality machined parts. Since NURBS curves are constructed piecewise using B-spline basis functions, the solution of curve trajectory points, the calculation of curve arc length, the derivation of NURBS curves, and the calculation of curvature surfaces have high computational loads. In particular, there is a nonlinear mapping relationship between the curve parameters and the arc length because of the segmented structure of the NURBS curve. 

Nie et al. [1] fully considered various constraints when planning the feedrate for the NURBS interpolator to improve the interpolation accuracy effectively. Liu et al. [2] generated smooth instructions that satisfy all the specified kinematic requirements. Wang et al. [3] introduced the pre-compensation-based theory to increase machining precision by lowering contour errors while maintaining federate. Cao et al. [4] suggested a feedrate direct interpolator (FDI), which can effectively acquire potentially precise interpolation parameters without using rounding or compensation techniques. Li et al. [5] employed the Sigmoid function for feedrate planning, considering both geometric and kinematic errors. Guo et al. [6] presented a federate scheduling method to overcome the NURBS curvature changing along the trajectory. Hu et al. [7] presented a brand-new feedrate scheduling approach based on an S-shape feedrate profile to improve interpolation accuracy. Wang et al. [8] introduced the cosines theorem in the NURBS interpolation algorithm to minimize feedrate fluctuations. Sang et al. [9] integrated morphological filtering with an S-shaped acc/dec profile for the NURBS interpolator in difficult regions. This resulted in more stable feedrates. Ji et al. [10] corrected the estimated parameters iteratively to achieve smaller velocity fluctuations and lower computation load. Jiang et al. [11] constrained the feedrate fluctuation by an iterative method. Zhao et al. [12] generated the smooth feedrate profile by improving the feed correction polynomial to reduce fluctuations. Zhang et al. [13] introduced a feedrate optimization method to achieve an ideal feedrate profile efficiently for the fastest machining efficiency. Ni et al. [14] designed a novel feedrate scheduling approach that maintains the machining efficiency of interpolation while keeping high motion smoothness. To minimize fluctuations, Zhong et al. [15] employed federate and acceleration lookahead operations. Ni et al. [15] employed a two-way control strategy to maintain NURBS interpolation accuracy under round-off errors. Cho and Kim [16] reduced the feedrate fluctuation using the compensation mechanism. Peng et al. [17] presented the improved Adams–Moulton (IAM) method to attain minimal feedrate fluctuation. Huang and Zhu [18] proposed a sinusoidal representation of the jerk profile for the parametric interpolator, thus making the feedrate profile more concise. These studies and analyses of the abovementioned literature indicate that direct interpolation of NURBS curves is a systematic project and obtaining high-speed and high-precision machining effects is essential.

Feedrate fluctuation during interpolation is unavoidable because of the improper mapping between the NURBS parameter and the arc length along the toolpath. The machining quality in NURBS interpolation is substantially influenced by feedrate fluctuation [19]. Many strategies have been put up to eliminate feedrate fluctuation successfully. For instance, UI (uniform increment) method [20] is the earliest direct approximation method. Although UI is the simplest method in this area, it may cause feedrate fluctuations. Taylor’s expansion (TE) based methods were widely utilized in the literature [21,22,23,24,25]. However, the unavoidable rounding-off error will cause feedrate fluctuation. An improved Taylor’s expansion (TE) method [26] was proposed to suppress feedrate fluctuation. The predictor–corrector interpolators (PCI) methods [27,28] can maintain feedrate fluctuations within specified tolerances after iteration while requiring variable iterations over successive sampling periods. Thus, the uncertain computation time caused by iteration is not conducive to interpolation stability and not suitable for high-speed real-time applications. Besides, there are remapping methods (RM) based on Hermite cubic splines [29], cubic polynomials [30], quintic splines [31], and seventh-order polynomials [32], which can reduce feedrate fluctuations. However, these methods require much storage space to store intermediate sub-polynomials and increase the computational load. Besides, the Steffensen-based iterative method [33] was also employed to refine the estimated initial parameter to reduce the feedrate fluctuation. However, this method requires continuous iterative and repeated complex parameter calculations, causing issues such as time-consuming calculations and enormous difficulties. Some scholars have proposed an arc-length compensation method [34], in which the command position is corrected by compensating for the arc and chord lengths in the parameter calculation process to reduce feedrate fluctuation. However, this method only considers the truncation error, and obtaining satisfactory results using the approximate method in the solution process is challenging. Super-resolution techniques [35,36] can be applied to the interpolated curves to enhance their resolution and improve their smoothness and curvature continuity. it is possible to improve the accuracy, precision, and overall quality of the interpolated curves, resulting in better performance and increased efficiency in various applications. However, super-resolution can bring a high computational load. 

To improve the accuracy of NURBS interpolator systems and minimize feedrate fluctuations during CNC machining, a two-level parameter compensation strategy is proposed. The corresponding feedrate fluctuation methods are designed for different regions according to the curvature characteristics, which can greatly reduce the feedrate fluctuation rate on the premise of ensuring real-time performance. The first-level parameter compensation is employed in the non-curvature sensitive area to control the feedrate fluctuation of the interpolation. In the curvature-sensitive area, the second-level parameter compensation is adopted after the first-level parameter compensation to overcome the problem of large federate fluctuations caused by large curvature. This strategy raises the tolerated time load and cannot affect the interpolator’s real-time performance. It generally utilizes the fact that feedrate changes are more noticeable in the curvature-sensitive area.

The novelty of the proposed two-level parameter compensation method lies in (1) proposing a two-level parameter compensation method specifically for NURBS interpolation to minimize feedrate fluctuation and improve machining quality, which utilizes the Taylor series expansion method in the first level and an iterative method without derivative calculations in the second level; (2) developing a novel approach for reducing feedrate fluctuation that considers the sensitivity of the curvature-sensitive areas, which can result in significant improvements in the quality of the final product; And (3) demonstrating the effectiveness of the proposed method through simulation studies and experiments, which show that the maximum feedrate fluctuation rate is below 0.01% with an average computational time of 360 us and that the proposed method outperforms other feedrate fluctuation elimination methods. Overall, the minimization of feedrate fluctuations is a novel contribution that can be applied to various applications, including those that require high precision and accuracy in motion control systems, such as 3D printing, CNC machining, and robotics.

The main contributions of this paper can be summarized as follows:

(1) We focused on the critical role of curvature in feedrate fluctuation, proposed the concept of the curvature-sensitive area, and divide the NURBS curve into curvature-sensitive areas and non-curvature-sensitive areas according to the curvature threshold to adapt to the curvature variations;

(2) A NURBS interpolator with minimum feedrate fluctuation based on two-level parameter compensation was proposed;

(3) The first-level parameter compensation based on Taylor series expansion was employed in the non-curvature-sensitive area; the second-level parameter compensation based on Secant method was employed in the curvature-sensitive area. Compared with the other four methods for eliminating feedrate fluctuation, the proposed method performed best and controlled the feedrate fluctuation within the standard range.

The remainder of this paper is structured as follows. Section 2 introduces the NURBS interpolation and feedrate fluctuation. Section 3 describes the feedrate fluctuation elimination method based on the two-level parameter compensation. The simulation results of applying the suggested method on NURBS curves with a butterfly shape are presented in Section 4. Finally, conclusions and future aspects are presented in Section 5.

## 2. NURBS Interpolation and Feedrate Fluctuation

### 2.1. The NURBS Curve and Parameter Discretization

NURBS curves are often expressed in the following rational fractional form [1]:(1)$$C(u)=\left[\begin{array}{c} x(u)\\ y(u)\\ z(u) \end{array}\right]=\frac{\sum_{i=0}^{n}N_{i,k}(u)\omega_{i}p_{i}}{\sum_{i=0}^{n}N_{i,k}(u)\omega_{i}}$$
where $P_{i}$ is the control point, $\omega_{i}$ is the weight of $P_{i}$, u is the non-uniform knot vector, and $N_{i,k}(u)$ is the *k*th-order B-spline basic function. 

Determining the curve parameter *u* for every interpolation period in the NURBS curve interpolation process is essential. The definition of the NURBS curve gives a deterministic mathematical expression for the free curve. Therefore, the key function of the NURBS interpolator is to discretize the curve parameter *u* and obtain the coordinates of interpolation point C(u) that meet the requirements of dynamic and motion characteristics.

The Taylor expansion approach effectively creates commands in general motion systems to discretize the curve parameter. By applying the Taylor expansion on the function u(t) at $t=t_{i}$ and ignoring the higher-order terms, the First-order Taylor expansion method can be expressed as
(2)$$u_{i+1}=u_{i}+{\frac{du}{dt}|}_{t=t_{i}}T$$

Additionally, the Second-order Taylor expansion method can be written as
(3)$$u_{i+1}=u_{i}+{\frac{du}{dt}|}_{t=t_{i}}T-{\frac{1}{2}\frac{d^{2}u}{dt^{2}}|}_{t=t_{i}}T^{2}$$

Then, the curve velocity $v(u_{i})$ can be derived as
(4)$$v(u_{i})={\left\|\frac{dC(u)}{dt}\right\|}_{u=u_{i}}={\left\|\frac{dC(u)}{du}\right\|}_{u=u_{i}}\cdot {\frac{du}{dt}|}_{t=t_{i}}$$

Since the curve parameter discretization by Taylor expansion approximation ignores the higher-order terms, there is a deviation between the discretized and actual values of *u*. Subsequently, this will cause a bias between the expected and actual arc lengths. Therefore, according to Equation (4), the bias of the arc length will cause a deviation between the interpolation and actual velocities. 

Besides, the CNC interpolation trajectory comprises micro-straight lines fed at $v(u_{i})$ per cycle. There is a gap between the feed micro-line and the planned arc trajectory, called the chord error. If the feedrate $v(u_{i})$ is inappropriate, it might become unsatisfactory. Throughout the interpolation procedure, the curve velocity $v(u_{i})$ must be modified adaptively depending on the curvature to maintain the chord error within a tolerance range. As shown in [37], the adaptive velocity corresponding to the maximum chord error can be obtained as:(5)$$v_{i}=\frac{2}{T}\cdot \sqrt{{\left(1/k_{i}\right)}^{2}-{\left(1/k_{i}-\delta_{\max}\right)}^{2}}$$
where $k_{i}$ is the curvature of the current interpolation point $v(u_{i})$. The interpolation feedrate cannot always be constant since the velocity is adaptively altered in response to curvature changes. Set the given maximum feedrate to *F*, then we can get
(6)$$v(u_{i})=\{\begin{array}{ll} F, & if\;v_{i}\geq F\\ v_{i}, & if\;v_{i}<F \end{array}$$

Substituting Equation (6) into Equation (2) yields the first-order adaptive-feedrate method (FAM):(7)$$\begin{array}{ll} u_{i+1} & =u_{i}+\frac{v(u_{i})}{\left\|C^{\prime }(u_{i})\right\|}T\\ & =\{\begin{array}{ll} u_{i}+\frac{FT}{\left\|C^{\prime }(u_{i})\right\|} & if\;v_{i}\geq F\\ u_{i}+\frac{v_{i}T}{\left\|C^{\prime }(u_{i})\right\|} & if\;v<F \end{array} \end{array}$$

Substituting Equation (6) into Equation (3) yields the second-order adaptive-feedrate method (SAM):(8)$$\begin{array}{l} u_{i+1}=u_{i}+\frac{v(u_{i})}{\left\|C^{\prime }(u_{i})\right\|}T-\frac{v^{2}(u_{i})C^{\prime }(u_{i})C^{\prime \prime }(u_{i})}{2{\left\|C^{\prime \prime }(u_{i})\right\|}^{4}}T^{2}\\ =\{\begin{array}{ll} u_{i}+\frac{FT}{\left\|C^{\prime }(u_{i})\right\|}-\frac{F^{2}T^{2}C^{\prime }(u_{i})C^{\prime \prime }(u_{i})}{{\left\|C^{\prime \prime }(u_{i})\right\|}^{4}} & if\;v_{i}\geq F\\ u_{i}+\frac{v_{i}T}{\left\|C^{\prime }(u_{i})\right\|}-\frac{{\left(v_{i}\right)}^{2}T^{2}C^{\prime }(u_{i})C^{\prime \prime }(u_{i})}{{\left\|C^{\prime \prime }(u_{i})\right\|}^{4}} & if\;v_{i}<F \end{array} \end{array}$$

The interpolation parameter $u_{pre,i+1}$ of the next point is first predicted according to the current adaptive velocity $v(u_{i})$ and the interpolation period by FAM or SAM. SAM has one more item than FAM, with higher calculation accuracy, while FAM should calculate the second-order derivative, which has higher calculation complexity.

### 2.2. Feedrate Fluctuation Analysis

Feedrate fluctuation, resulting from a nonlinear correspondence between the curve parameters and the arc length displacement, is the difference between the desired and actual feed rates. The actual feedrate during the interpolation process can be expressed as [34]
(9)$$\underset{actual\;feedrate\;command}{\underset{︸}{\dot{s}=(ds/dt)}}=\underset{feedrate\;\mathrm{flucuation}}{\underset{︸}{\left\|C^{\prime }(u)\right\|\cdot (d\hat{u}/ds_{d})}}\underset{desired\;\mathrm{feedrate}}{\underset{︸}{\left(ds_{d}/dt\right)}}$$
where *s* is the actual arc length, $C^{\prime }(u)$ is the derivative of the NURBS curve, ‖ ‖ is the Euclidean norm, $\hat{u}$ is the predicted parameter, and $s_{d}$ is the desired arc length.

There is no accurate analytical expression for the NURBS curve’s parameters and arc length because of their unique definitions in Equation (1). The acquisition of the predicted parameter $\hat{u}$ generally adopts an approximate method (such as FAM or SAM). Because of the deviation of the estimated parameter $\hat{u}$ from the actual value, the arc displacement along the curved path is inaccurately mapped to the spline parameters, causing the term $\left\|C^{\prime }(u)\right\|\cdot (d\hat{u}/ds_{d})$ not to be equal to 1. Consequently, the inequality $\underset{actual\;feedrate\;command}{\underset{︸}{\dot{s}}}\neq \underset{desired\;\mathrm{feedrate}}{\underset{︸}{\left(ds_{d}/dt\right)}}$ holds and feedrate fluctuations follow.

According to Figure 1, A and B represent the actual *i*th point and the predicted (*i* + 1)th point. $\overset{⌢}{AB}$ and $\bar{AB}$ are the arc and chord lengths, respectively. The arc length between A and B is $S_{i+1}$, and the chord length $\bar{AB}$ is $l_{i+1}$. When the arc length is tiny enough, a circle with the same curvature can approximate it. $R_{i}$ is the curvature radius at point A.

Interpolation feedrate can be obtained as
(10)$$v_{i+1}=\frac{L_{i+1}}{T}$$

The theoretical feedrate can be calculated as
(11)$${v^{\prime }}_{i+1}=\frac{S_{i+1}}{T}$$

For the path with nonzero curvatures, as depicted in Figure 1, the expression $l_{i+1}\neq S_{i+1}$ holds, which results in $v_{i+1}\neq {v^{\prime }}_{i+1}$. Consequently, feedrate fluctuations during the NURBS curve interpolation may result from discrepancies between the actual and predicted trajectories.

Thus, the feedrate fluctuations rate of the (*i* + 1)th interpolation point can be defined as
(12)$$\varepsilon_{i+1}=\frac{{v^{\prime }}_{i+1}-v_{i+1}}{v_{i+1}}$$

It can be rewritten as
(13)$$\varepsilon_{i+1}=\frac{S_{i+1}-l_{i+1}}{l_{i+1}}$$

From Figure 1, according to the relationship between arc length and radian, it can be obtained as
(14)$$\frac{S_{i+1}}{2}=R_{i}\cdot \theta =R_{i}\cdot \arcsin\left(\frac{l_{i+1}}{2R_{i}}\right)$$

*R_i_* is the curvature radius at point A

In (14), the arcsine term can be roughly calculated using the Maclaurin series, which is written as
(15)$$\arcsin\left(\frac{l_{i+1}}{2R_{i}}\right)\approx \frac{l_{i+1}}{2R_{i}}+\frac{1}{6}{\left(\frac{l_{i+1}}{2R_{i}}\right)}^{3}$$

Substituting Equation (15) into Equation (14) yields
(16)$$S_{i+1}=2\left(R_{i}\cdot \arcsin\left(\frac{l_{i+1}}{2R_{i}}\right)\right)=2R_{i}\left(\frac{l_{i+1}}{2R_{i}}+\frac{1}{6}{\left(\frac{l_{i+1}}{2R_{i}}\right)}^{3}\right)=l_{i+1}+\frac{{\left(l_{i+1}\right)}^{3}}{24R_{i}{}^{2}}$$

Substituting Equation (16) into Equation (13) gives
(17)$$\varepsilon_{i+1}=\frac{S_{i+1}-l_{i+1}}{l_{i+1}}=\frac{l_{i+1}+\frac{{\left(l_{i}\right)}^{3}}{24{\left(R_{i}\right)}^{2}}-l_{i+1}}{l_{i+1}}=\frac{{\left(l_{i}\right)}^{3}}{24{\left(R_{i}\right)}^{2}l_{i+1}}$$

Equation (17) shows that the curvature radius change directly determines the feedrate fluctuation’s magnitude. 

As the reciprocal of the curvature equals the curvature radius, $R_{i}=\frac{1}{k(u_{i})}R$ holds, where $k(u_{i})$ is the curvature obtained as follows:(18)$$k(u_{i})=\frac{C^{\prime }(u_{i})\times C^{\prime \prime }(u_{i})}{{\left|C^{\prime }(u_{i})\right|}^{3}}$$
where the first and second-order derivatives of the current point are denoted by $C^{\prime }(u_{i})$ and $C^{\prime \prime }(u_{i})$, respectively.

Equation (17) can be rewritten as
(19)$$\varepsilon_{i+1}=\frac{{\left(l_{i}\right)}^{3}}{24{\left(\frac{1}{k\left(u_{i}\right)}\right)}^{2}l_{i+1}}=\frac{{\left(l_{i}\right)}^{3}{\left(k\left(u_{i}\right)\right)}^{2}}{24l_{i+1}}$$

Under the same actual interpolation trajectory length l, it can be concluded from Equation (19) that greater curvature leads to more significant feedrate fluctuations. After eliminating the feedrate fluctuations, it is necessary to take corresponding measures combined with the curvature properties.

## 3. Two-Level Parameter Compensation

In this research, we aimed to address the issue of feedrate fluctuations that occur during NURBS interpolation. Such fluctuations can significantly impact the quality, accuracy, and efficiency of Computer Numerical Control (CNC) machining processes. Although various methods have been proposed in previous studies to reduce these fluctuations, most of them involve complex calculations and are not conducive to real-time and high-precision machining applications. Hence, we proposed a new two-level parameter compensation method that effectively minimizes feedrate fluctuations, considering the curvature-sensitive areas in the NURBS curve.

### 3.1. Feedrate Fluctuation Elimination Strategy

In order to balance efficiency and effectiveness, a two-level parameter compensation method is suggested to avoid feedrate fluctuations. Figure 2 shows the schematic diagram of two-level parameter compensation for the (*i* + 1)th interpolation point. Firstly, to address the federate fluctuation in non-curvature sensitive areas with low computational cost, first-level parameter compensation is performed. The compensation is achieved through the Taylor series expansion method, ensuring that the chord trajectory of the new interpolation point after compensation closely matches the original arc trajectory. Secondly, although the first-level parameter compensation reduces truncation error in curvature-sensitive areas, significant feedrate fluctuations can still occur. To further address this issue, second-level parameter compensation is performed using an iterative method that does not require derivative calculations.

As shown in Figure 2, initially, parameters such as $u_{i}$ and curvature $k(u_{i})$ of the current interpolation point, command feedrate *F*, the maximum chord error $\delta_{\max}$, curvature threshold $k_{threshold}$, and the feedrate fluctuation tolerance ε are input into the two-level parameter compensation module. Then, the first-level parameter compensation is performed. In this process, the adaptive velocity of the next interpolation point is calculated using the adaptive velocity approach described in Equation (5) and is restrained under the command feedrate constraint according to Equation (6). The estimated parameters $u_{pre,i+1}$ of the next interpolation point are obtained by using the FAM or SAM described in Equations (7) and (8). As the numerical approximation method is utilized to obtain the estimated parameters, there are inevitable discrepancies between these parameters and the actual ones. Consequently, the arc displacement along the curved path is inaccurately mapped to the spline parameters, leading to fluctuations in the federate. In this paper, we provided a theoretical derivation process for compensating for the problem of feedrate fluctuation caused by parameter estimation deviation. More details can be found in Section 3.2. Therefore, we can obtain the parameter compensation step size $\Delta u_{i+1}$ according to the method called first-level compensation in this paper. In this way, the first-level parameter compensation is completed and we can update the new predicted parameter through the formula $u_{f,i+1}=u_{pre,i+1}+\Delta u_{i+1}$.

Greater curvature leads to more prominent fluctuation in feedrate. Even after eliminating the feedrate variation through the first-level parameter compensation in the curvature-sensitive area, there is still a pressing issue of substantial feedrate fluctuations. This paper presents a second-level parameter compensation method that is combined with curvature characteristics to address this problem. As shown in Figure 2, after completing the first-level parameter compensation, it is important to check whether the next interpolation point is situated in the curvature-sensitive area. If the point is located in the non-curvature-sensitive area, the NURBS interpolator will proceed with the calculation process, which involves obtaining the interpolation position based on the parameters. Once this is done, the feed system will execute the interpolation action to complete the feed process. If the point falls within the curvature-sensitive area, then proceed to the second-level parameter compensation based on the Secant iteration method. First, initialize the parameters, including the number of iterations, $u_{k}$ and $u_{k-1}$, etc. In each iteration, a new compensated parameter $u_{k+1}$ is updated according to the formula $u_{k+1}=u_{k}-\frac{f(u_{k})\left(u_{k-1}-u_{k}\right)}{f(u_{k-1})-f(u_{k})}$. Then, the estimated feedrate $v_{k+1}$ is obtained by the ratio of the distance between two points $u_{k+1}$ and $u_{i}$, which are the new compensated interpolation point and the previous interpolation point, to the interpolation period. After that, the new feedrate fluctuation is gained and is compared with the set fluctuation tolerance ε. If the feedrate fluctuation is still higher than the set tolerance, the number of iterations k will update and the next update iteration process proceed. If the current parameter compensation results in the feedrate fluctuation being within the set tolerance and meets the accuracy condition, then, the second-level compensation will end. Next, the NURBS interpolator will solve the curve point coordinates and perform the feed motion. The specific process of second-level parameter compensation is detailed in Section 3.3. After two-level parameter compensation, the feedrate fluctuation of the estimated point can be well eliminated, and the quality of the processed surface can be better guaranteed. 

High curvature regions are the curve parts that exceed the curvature threshold in the second-level parameter compensation. The NURBS curve is firstly divided into low and high curvature regions by the curvature threshold. The curve’s curvature threshold is determined by considering dynamic limitations and geometric properties. Taking F as the maximum command feedrate, and $\delta_{\max}$ as the chord error tolerance, the curvature threshold can be determined from Equation (5) as
(20)$$k_{threshold}=\frac{8\delta_{\max}}{F^{2}T^{2}+\delta_{\max}^{2}}$$

Then, the curve is divided into numerous sections based on the curvature threshold. The NURBS curve’s high and low-curvature areas are defined as the portion with curvature larger and less than the threshold, respectively. It can be seen from the curvature characteristics that when the interpolation point is located in the low curvature area, the feedrate fluctuation is slight, and only the first-level parameter compensation is required. In contrast, when the interpolation point is located in the high curvature area, the feedrate fluctuation is significant, and it is necessary to perform the second-level parameter compensation after the first-level parameter compensation.

### 3.2. The First-Level Parameter Compensation

The main factor causing feedrate fluctuation is that the synthesis trajectory is a short straight line instead of a circular arc. In order to make the chord trajectory of the new interpolation point after compensation as similar to the original arc trajectory as possible, the first-level parameter compensation was performed, as shown in Figure 1, where C is the new point of point B after the first-level compensation. 

The parameters of B and C were set as $u_{pre,i+1}$$u_{f,i+1}$. The parameter compensation incremental value of point B was set to $\Delta u_{i+1}$. Now, we have
(21)$$\Delta u_{i+1}=u_{f,i+1}-u_{pre,i+1}$$
where $u_{pre,i+1}$ is the predicted parameter of the next interpolation point by FAM or SAM.

Here, the first-order Taylor series expansion technique establishes a model between the interpolation point and the compensation value [38].
(22)$$C(u_{f,i+1})=C(u_{pre,i+1})+C^{\prime }(u_{pre,i+1})\Delta u_{i+1}$$
where $C^{\prime }(u_{pre,i+1})$ is the first-order derivative of $C(u_{pre,i+1})$.

After the first-level parameter compensation, it is assumed that there is no feedrate fluctuation, and $\overset{⌢}{AB}=\bar{AB}$ holds. This also means that the desired feedrate should equal the actual feedrate. It yields
(23)$$\frac{S_{i+1}}{T}=\frac{\left\|C(u_{f,i+1})-C(u_{i})\right\|}{T}$$

Equation (23) can be rewritten by
(24)$$v_{i+1}=\frac{\left\|C(u_{f,i+1})-C(u_{i})\right\|}{T}$$

By substituting (22) with (24), we obtain
(25)$$\left\|C(u_{pre,i+1})+C^{\prime }(u_{pre,i+1})\Delta u_{i+1}-C(u_{i})\right\|=v_{i+1}T$$

The quadratic equation for Δ*u*_*i*+1_ can be obtained by squaring both sides of Equation (25).
(26)$${\left\|C^{\prime }(u_{pre,i+1})\right\|}^{2}\Delta u_{i+1}^{2}+2(C^{\prime }(u_{pre,i+1}),C(u_{pre,i+1})-C(u_{i}))\Delta u_{i+1}+{\left\|C(u_{pre,i+1})-C(u_{i})\right\|}^{2}-v_{i+1}^{2}T^{2}=0$$

Then, it can be rewritten as
(27)$$a\Delta u_{i+1}^{2}+b\Delta u_{i+1}+c=0$$
where *a*, *b*, and *c* are listed as follows:(28)$$\{\begin{array}{c} a={\left\|C^{\prime }(u_{pre,i+1})\right\|}^{2}\\ b=2(C^{\prime }(u_{pre,i+1}),C(u_{pre,i+1})-C(u_{i}))\\ c={\left\|C(u_{pre,i+1})-C(u_{i})\right\|}^{2}-v_{i+1}^{2}T^{2} \end{array}$$

Two roots of Equation (27) can be obtained as
(29)$$\Delta u_{i+1,1}=\frac{-b+\sqrt{b^{2}-4ac}}{2a}$$
and
(30)$$\Delta u_{i+1,2}=\frac{-b-\sqrt{b^{2}-4ac}}{2a}$$

Since $c={\left\|C(u_{f,i+1})-C(u_{i})\right\|}^{2}-v_{i+1}^{2}T^{2}\approx 0$, one of the roots turns out to be close to zero, and the other root is $\Delta u_{i+1,2}=-\frac{b}{a}$. Therefore, the parameter compensation should be slight, and the parameter compensation is taken to be the value of $\Delta u_{i+1,1}$. 

### 3.3. The Second-Level Parameter Compensation

The FAM or SAM based on Taylor expansion employed in the first-level compensation has truncation errors, which cannot be ignored in high curvature areas, and will still cause feedrate fluctuations. 

Suppose a point in the high curvature region with feedrate $v_{i+1}$. Assuming there are no feedrate fluctuations on this point, it just requires finding parameter u to satisfy f(u)=0 in Formula (31).
(31)$$F(u)=\left\|C(u)-C(u_{i})\right\|-v_{i}T=0$$
where f(u) represents the difference between the linear distance of adjacent interpolation points and the actual interpolation trajectory length. When f(u) equals 0, there is no feedrate fluctuation. The Secant method is introduced to solve (31), an iterative method without calculating the derivatives. Suppose the feedrate fluctuation tolerance is *ε*. The pseudocode representation of the method is shown in Algorithm 1.
**Algorithm 1.** Secant-Method Based Parameter Compensation Input: $u_{i}$, $v_{i}$, $u_{\mathrm{pre},i+1}$, $u_{f,i+1}$, εOutput: $u_{i+1}$    1: initialize: Set *k* = 1, $u_{k}=u_{f,i+1}$, $u_{k-1}=u_{\mathrm{pre},i+1}$    2: do     3:                        $u_{k+1}=u_{k}-\frac{f(u_{k})(u_{k-1}-u_{k})}{f(u_{k-1})-f(u_{k})}$
4:                          $v_{k+1}=\frac{\left\|C(u_{k+1})-C(u_{i})\right\|}{T}$    5:   until $\frac{\left|v_{i+1}-v_{k+1}\right|}{v_{i+1}}\leq \varepsilon$    6: $u_{i+1}=u_{k}$


In Algorithm 1, the input parameters include parameter $u_{i}$, the desired feedrate $v_{i+1}$, the predicted parameter $u_{pre,i+1}$, the first-level parameter compensation $u_{f,i+1}$, and the feedrate fluctuation tolerance ε. $u_{k+1}$ represents the parameter compensation of Equation (31) at the (*k* + 1)th iteration, and $v_{k+1}$ indicates the actual feedrate according to parameter $u_{k+1}$. If the feedrate fluctuation is under the upper bound ε, then $u_{k+1}$ is set to the desired value.

### 3.4. The Order of Convergence for the Second-Level Parameter Compensation

Let $e_{k}$ be an error in the *k*th iteration approximation. The parameter compensations $u_{k}$ and $u_{k-1}$ can be written as
(32)$$\left\{ \begin{array}{l} u_{k}=u+e_{k}\\ u_{k-1}=u+e_{k-1} \end{array}\right.$$

Substituting Equation (32) into Equation (31) and expanding with the Taylor series, we get
(33)$$\left\{ \begin{array}{l} f(u_{k})=f(u+e_{k})=f(u)+e_{k}f^{\prime }(u)+\frac{e_{k}^{2}}{2}f^{\prime \prime }(u)\;\\ f(u_{k-1})=f(u+e_{k-1})=f(u)+e_{k-1}f^{\prime }(u)+\;\frac{e_{k-1}^{2}}{2}f^{\prime \prime }(u) \end{array}\right.$$

Since f(u)=0, Equation (33) can be rewritten as
(34)$$\left\{ \begin{array}{l} f(u_{k})=e_{k}f^{\prime }(u)+\frac{e_{k}^{2}}{2}f^{\prime \prime }(u)\;\\ f(u_{k-1})=e_{k-1}f^{\prime }(u)+\frac{e_{k-1}^{2}}{2}f^{\prime \prime }(u)\; \end{array}\right.$$

Now, the following formula can be obtained
(35)$$f(u_{k})-f(u_{k-1})=(e_{k}-e_{k-1})f^{\prime }(u)+\frac{1}{2}(e_{k}^{2}-e_{k-1}^{2})f^{\prime \prime }(u)\;$$

The second-level iteration parameter compensation can be described as
(36)$$\begin{array}{ll} u_{k+1} & =u_{k}-\frac{f(u_{k})\left(u_{k-1}-u_{k}\right)}{f(u_{k-1})-f(u_{k})}\\ & =\frac{u_{k-1}f(u_{k})-u_{k}f(u_{k-1})}{f(u_{k})-f(u_{k-1})} \end{array}$$

Substituting Equations (34) and (35) into Equation (36) gives
(37)$$\begin{array}{ll} e_{k+1}+u & =\frac{(e_{k-1}+u)\left(e_{k}f^{\prime }(u)+\frac{e_{k}^{2}}{2}f^{\prime \prime }(u)\right)-(e_{k}+u)\left(e_{k-1}f^{\prime }(u)+\frac{e_{k-1}^{2}}{2}f^{\prime \prime }(u)\right)}{\left(e_{k}-e_{k-1})f^{\prime }(u)+\frac{1}{2}(e_{k}^{2}-e_{k-1}^{2})f^{\prime \prime }(u\right)}\\ & =\frac{(e_{k}-e_{k-1})\left[uf^{\prime }(u)+\frac{u}{2}f^{\prime \prime }(u)(e_{k}-e_{k-1})+\frac{e_{k}e_{k-1}}{2}f^{\prime \prime }(u)\right]}{(e_{k}-e_{k-1})\left[f^{\prime }(u)+\frac{1}{2}(e_{k}+e_{k-1})f^{\prime \prime }(u)\right]} \end{array}$$

Equation (37) can be rewritten as
(38)$$\begin{array}{ll} e_{k+1} & =-u+\frac{uf^{\prime }(u)+\frac{u}{2}f^{\prime \prime }(u)(e_{k}-e_{k-1})+\frac{e_{k}e_{k-1}}{2}f^{\prime \prime }(u)}{f^{\prime }(u)+\frac{1}{2}(e_{k}+e_{k-1})f^{\prime \prime }(u)}\\ & =\frac{\frac{e_{k}e_{k-1}}{2}f^{\prime }(u)}{f^{\prime }(u)\left[1+\frac{1}{2}(e_{k}+e_{k-1})\frac{f^{\prime \prime }(u)}{f^{\prime }(u)}\right]}\\ & =\frac{e_{k}e_{k-1}}{2}\cdot \frac{f^{\prime \prime }(u)}{f^{\prime }(u)}{\left[1+\frac{e_{k}+e_{k-1}}{2}\frac{f^{\prime \prime }(u)}{f^{\prime }(u)}\right]}^{-1} \end{array}$$

By binomial theorem, Equation (38) can be rewritten as
(39)$$e_{k+1}=\frac{e_{k}e_{k-1}}{2}\cdot \frac{f^{\prime \prime }(u)}{f^{\prime }(u)}\left[1-\frac{e_{k}+e_{k-1}}{2}\frac{f^{\prime \prime }(u)}{f^{\prime }(u)}\right]$$

Neglecting powers of $e_{k}$ and $e_{k-1}$ yields
(40)$$e_{k+1}\approx \frac{e_{k}e_{k-1}}{2}\frac{f^{\prime \prime }(u)}{f^{\prime }(u)}$$
(41)$$or\;e_{k+1}\propto e_{k}e_{k-1}$$

If *p* is the order of convergence of iterations, then
(42)$$e_{k+1}\propto e_{k}^{p}$$
(43)$$\mathrm{Or}\;e_{k}\propto e_{k-1}^{p}$$

According to Equations (41) and (42), we have
(44)$$e_{k}^{p}\propto e_{k}e_{k-1}$$

From Equation (43), we have
(45)$$\begin{array}{l} e_{k}^{p}\propto e_{k-1}^{p}e_{k-1}\\ e_{k}\propto e_{k-1}^{p+1} \end{array}$$

And then
(46)$$e_{k}\propto e_{k-1}^{p+1/p}$$

From Equations (43) and (46), we can obtain
(47)$$p=\frac{p+1}{p}\;\mathrm{or}\;p^{2}-p-1=0$$

The roots of Equation (47) are
(48)$$p=\frac{1\pm \sqrt{1+4}}{2}=\frac{1\pm \sqrt{5}}{2}$$

Taking the positive sign, we have $p=\frac{1+\sqrt{5}}{2}\approx 1.618$.

The order of convergence of the second-level parameter compensation is 1.618, which is superlinear. 

## 4. Simulation Results

### 4.1. The Simulation Parameters and Methods

In order to demonstrate the feasibility and effectiveness of the supposed approach, the typical butterfly-shaped NURBS curve is simulated. The typical butterfly-shaped curve and its control polygon are shown in Figure 3. In this simulation, the command feedrate, maximum chord error, and interpolation period are set to 0.1 mm/ms, 0.001 mm, and 2 ms, respectively. 

Figure 4 depicts the curvature of the butterfly-shaped curve. The curvature threshold is 0.19998, calculated by the given parameters according to Equation (20), as shown by the red line in Figure 4. As shown in Figure 4, most regions are below the curvature threshold, while there are also several regions above the curvature threshold, which are high curvature areas that require performing second-level parameter compensation after the first-level parameter compensation. In Figure 4, the part with the parameter range of 0.30–0.40 is selected and magnified (the area indicated by the arrow in Figure 4), which can be seen more clearly.

This paper performed simulation experiments on the first-level and second-level parameter compensations under the FAM and SAM. The horizontal axis shows the normalized parameter vector and its value range is 0~1. The vertical axis shows the feedrate fluctuation rate as a percentage. 

### 4.2. Simulation Analysis

In Figure 5, the NURBS curve interpolation adopts the FAM, which does not consider the deviation between the estimated and theoretical interpolation trajectories. Figure 5 shows that this method gives significant feedrate fluctuations, and the maximum feedrate fluctuation rate reaches 18.39%. Such high feedrate fluctuations will directly degrade the interpolation accuracy and surface quality.

In Figure 6, the first-level parameter compensation is performed on FAM. For the parameter estimated by the FAM in Equation (7), the compensation increment is calculated according to Equation (29), and the compensation increment is added to update the next point parameters. Figure 6 shows that the FAM algorithm’s feedrate fluctuation range is significantly reduced after the first-level parameter compensation, and the maximum fluctuation rate is reduced from 18.39% to 1.2%. The feedrate fluctuation after the first-level parameter compensation in FAM is 93.5% lower than without parameter compensation. However, Figure 6 shows that most feedrate fluctuations are less than 0.2%, and the area with significant feedrate fluctuations corresponds to the curvature-sensitive area shown in Figure 4.

Since the feedrate fluctuation is still relatively significant after the first-level parameter compensation in the curvature-sensitive areas, the second-level parameter compensation method is adopted. Figure 7 shows the feedrate fluctuation of the FAM with two-level parameter compensation. This paper selected the maximum number of iterations as 1~5 in the second-level parameter compensation for real-time performance and investigated the feedrate fluctuation under the fixed iteration period. Figure 7 shows the feedrate fluctuation reduction under different iterations. When the iteration parameter was set to 1, the maximum feedrate fluctuation rate was reduced to 0.5036%, 58.03% lower than that of the FAM with the first-level parameter compensation. When the iteration parameter was set to 5, the feedrate fluctuation rate in the curvature-sensitive area continued to decrease, falling below 0.1%.

In order to further eliminate the feedrate fluctuation rate, this paper also simulated two-level parameter compensation of the SAM, as shown in Figure 8, Figure 9 and Figure 10. Figure 8 depicts the SAM’s feedrate fluctuation without parameter compensation. Since SAM has one more second-order derivative than FAM, its truncation error is smaller than that of FAM, and its estimated parameter of the interpolation point is closer to the theoretical value. Therefore, the feedrate fluctuation rate in the SAM is lower than in the FAM. Nevertheless, the maximum feedrate fluctuation rate of the SAM still reached 6.32%, which cannot meet the high precision machining requirements. 

Figure 9 depicts the SAM’s feedrate fluctuation with the first-level parameter compensation. Similar to the simulation results in Figure 6, the feedrate fluctuation rate was significantly reduced (from a maximum of 1.2% to 0.178%, a drop of 85.17%) after adopting the first-level parameter compensation. The main reason is that the parameters estimated by the SAM in Equation (8) are compensated according to Equation (29), so the estimated parameters were closer to the theoretical ones.

However, Figure 9 shows that corresponding to the curvature-sensitive areas in Figure 4, the feedrate fluctuations were more significant than other areas, which still cannot meet the high-quality machining requirements. The main reason is that the first-level compensation has a truncation error, and the length of the interpolation trajectory is minimal in the curvature-sensitive area. Thus, the effect of truncation error on feedrate fluctuation cannot be ignored.

Therefore, the secondary second-level parameter compensation is adopted in SAM, and the feedrate fluctuation rate is shown in Figure 10. Similar to the second-level parameter compensation of the FAM, the maximum number of iterations was specified as 1~5 in the second-level parameter compensation, and the feedrate fluctuation under the fixed iteration period was investigated. As shown in Figure 10, when the iteration parameter was set to 1, the maximum feedrate fluctuation rate after parameter compensation for the curvature-sensitive area was reduced to 0.0679%, 61.85% lower than the SAM with the first-level parameter compensation. When the iteration parameter was set to 5, the feedrate fluctuation rate in the curvature-sensitive area decreased, falling below 0.01%. Even if the iteration parameter was set to 1 in the SAM with the two-level parameter compensation, the feedrate fluctuation error was sufficient for high-precision machining compared with the standard range of 0.001–0.1% [8].

Table 1 shows a comparison of the performance of parameter compensation interpolation algorithms. The FAM has a maximum feedrate fluctuation rate of 18.39%. The maximum federate fluctuation rate of the FAM with the first-level parameter compensation was 93.5% lower than the FAM. When using the two-level parameter compensation and setting the iteration parameter to one, the maximum feedrate fluctuation rate was 0.5, 97.3% less than the FAM. When the iteration parameter was set to 5, the feedrate fluctuation rate was 0.1%, 99.5% lower than the FAM. Correspondingly, in terms of the sum of feedrate fluctuations in the whole interpolation process, the FAM with the first-level parameter compensation was 99.19% lower than that without compensation, and the two-level parameter compensation was reduced by 99.56% and 99.71% when the iteration parameters were set to 1 and 5, respectively.

Besides, for the SAM, the proposed two-level parameter compensation also provided excellent performance, as shown in Table 1. The maximum rate of feedrate fluctuations obtained with the SAM was 6.32%, while it was 0.18% for the SAM with the first-level parameter compensation, 97.2% lower than the SAM. When the two-level parameter compensation was adopted and the iteration parameter was set to 1, the maximum rate of feedrate fluctuation was 0.07%, 98.9% lower than that of the SAM. When the iteration parameter was set to 5, the maximum rate of feedrate fluctuation was 0.01%, 99.8% lower than the SAM. Similarly, the maximum rate of feedrate fluctuations obtained with the SAM with the first-level parameter compensation was 97.10% lower than without compensation, and the two-level parameter compensation reduced it by 97.88% and 98.00% when the iteration parameter was set to 1 and 5, respectively. At the same time, in the case of horizontal comparison, when the two-level parameter compensation was adopted and the number of iterations was set to 5, SAM’s maximum rate of feedrate fluctuation was only one-tenth that of FAM. Under the SAM with the two-layer parameter compensation, even if the iteration parameter was set to 1, the feedrate fluctuation was sufficient for high-precision machining.

Furthermore, the FAM and SAM without parameter compensation consumed the least computational time while generating the worst feedrate fluctuation rate. The time required for the first-level parameter compensation to compute the NURBS curve’s first derivative was doubled. However, the feedrate fluctuation rate decreased by more than 93% after adopting FAM and SAM. The second-level parameter compensation does not require calculating the first derivative of the NURBS curve while adding some simple calculations with no discernible time increase. The feedrate fluctuation control was further improved after adopting FAM and SAM. The SAM with the two-level parameter compensation (iteration is 5) consumed most time than the other methods and could reach the maximum rate of feedrate fluctuation below 0.01%. The proposed two-level parameter compensation method provided the best feedrate accuracy compared with the other methods. Furthermore, the proposed method has a computational time of about 350 us for FAM and 360 us for SAM, which is acceptable for the specified interpolation period (2 ms). As a result, the proposed method’s real-time performance can meet the requirements of typical computer numerical control (CNC) systems.

Table 2 compares the following four interpolation methods: the proposed SAM with two-level parameter compensation, the IRF method [26], Fast NURBS (fifth-degree ILF) [29], the TFSS method [12], and the OFB method [37]. The five methods had maximum federate fluctuation rates of 0.01%, 2%, 0.025%, 0.25%, and 0.027%, respectively. The proposed method had the smallest feedrate fluctuation among the four methods. The feedrate fluctuations of the other four methods were significantly higher than the proposed method, which were 1.99 × 10^3^%, 1.5 × 10^2^%, 2.4 × 10^3^%, and 1.7 × 10^2^%.

### 4.3. Physical Mechanisms Analysis

The Taylor expansion approximation method disregards higher-order terms, inducing truncation errors, and discrepancies between estimated and theoretical interpolation points. As a result, it creates feedrate fluctuations that exceed the required range for high-speed and high-precision machining. This paper suggests a first-level parameter compensation method that establishes a mathematical analytical solution for the truncation error compensation and compensates for the error with a step increment. The simulation results indicated that utilizing a first-level parameter compensation for the first-order and second-order Taylor expansion methods can decrease feedrate fluctuation to less than 0.02% in non-curvature sensitive areas. However, fluctuations in curvature-sensitive areas remain significant because of the continuous impact of curvature on NURBS interpolation. To solve this issue, the paper suggests the second-level parameter compensation to further limit the impact of curvature on feedrate fluctuation. The Secant method was introduced to iteratively regulate feedrate fluctuation within the fluctuation tolerance. The simulation results also demonstrated that feedrate fluctuation can be effectively managed within 0.01% after the second-level parameter compensation is implemented.

## 5. Conclusions

This paper suggests a two-level parameter compensation method for the NURBS interpolator to minimize feedrate fluctuation. First, the Taylor series expansion method was utilized in the first-level parameter compensation to obtain precise parameters to make the chord trajectory of the new interpolation point after compensation as close to the original arc trajectory as possible, eliminating federate fluctuation in non-curvature-sensitive areas with low calculation cost. Second, the second-level parameter compensation was performed based on the Secant method to overcome the truncation error of the first-level parameter compensation, an iterative method without derivative calculation requirement. Finally, the butterfly-shaped NURBS curves were simulated. The simulation results indicated that the maximum feedrate fluctuation rate was below 0.01% with an average computational time of 360 us in SAM with two-level parameter compensation, which is 99.8% lower than the SAM without parameter compensation. The proposed method performed best among the four feedrate fluctuation elimination methods. Compared with the standard range, the simulation demonstrated its feasibility and effectiveness for high-precision and real-time machining. However, this paper still has limitations, for example, the effectiveness of the proposed method in minimizing feedrate fluctuations during actual CNC machining needs to be tested experimentally. Future research will investigate the potential of machine learning and artificial intelligence algorithms in developing adaptive and self-optimizing feedrate control systems for CNC machining.

## Figures and Tables

**Figure 1 sensors-23-03789-f001:**
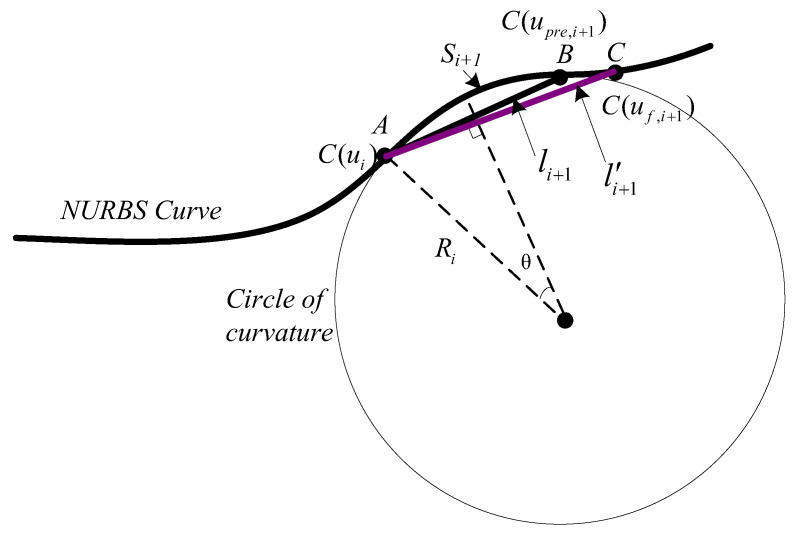
NURBS curve interpolation trajectory.

**Figure 2 sensors-23-03789-f002:**
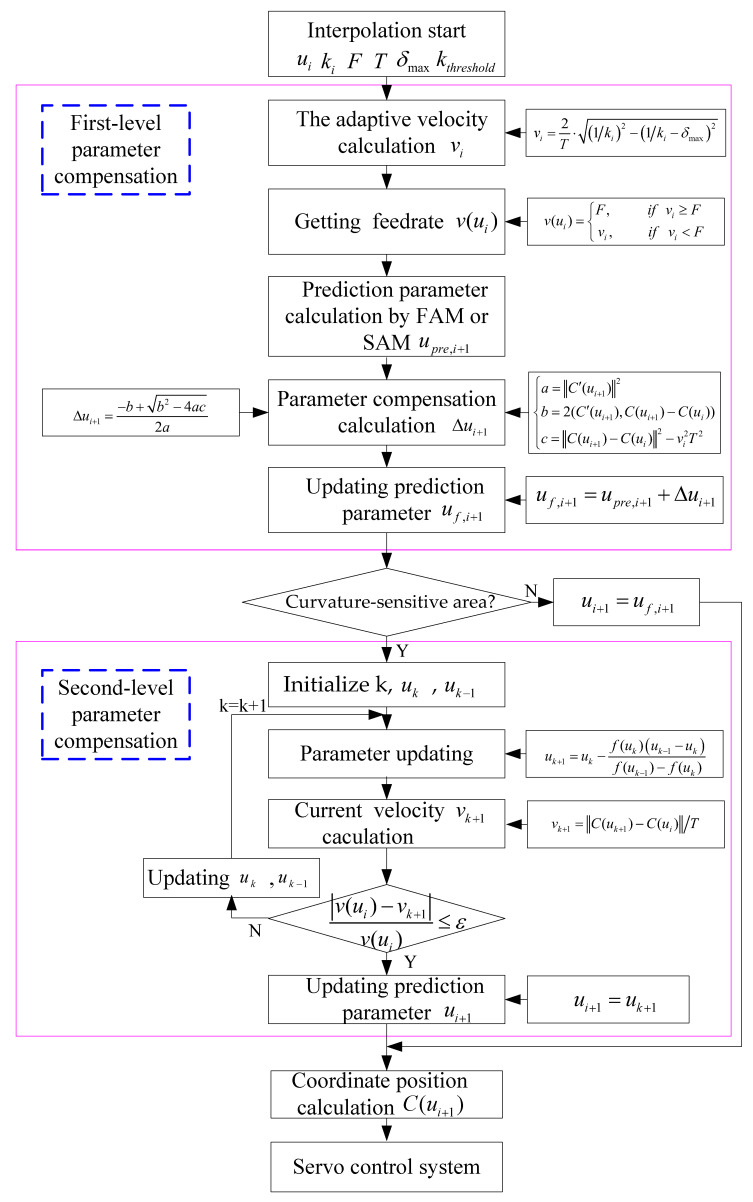
The flowchart of the parameter compensation.

**Figure 3 sensors-23-03789-f003:**
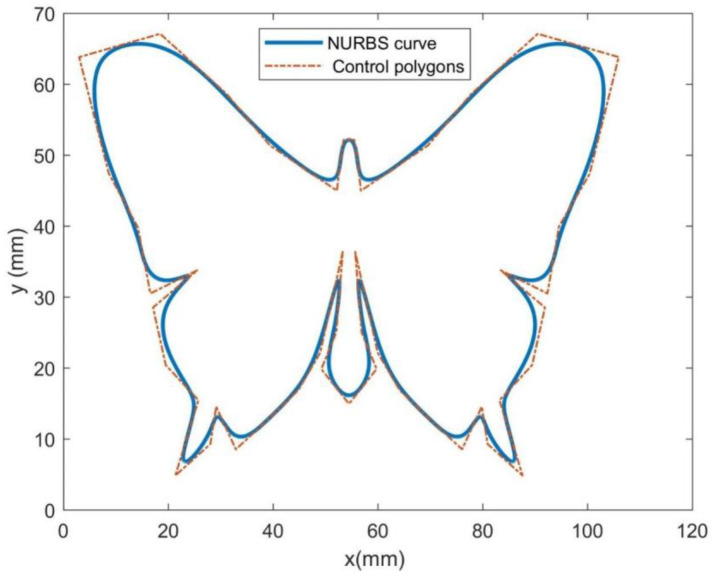
Butterfly-shaped NURBS curve.

**Figure 4 sensors-23-03789-f004:**
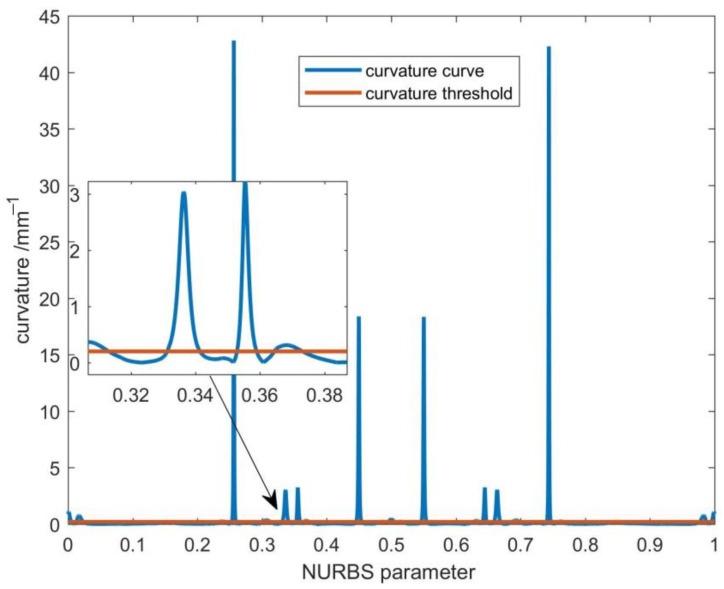
Curvature of the NURBS curve and the curvature threshold.

**Figure 5 sensors-23-03789-f005:**
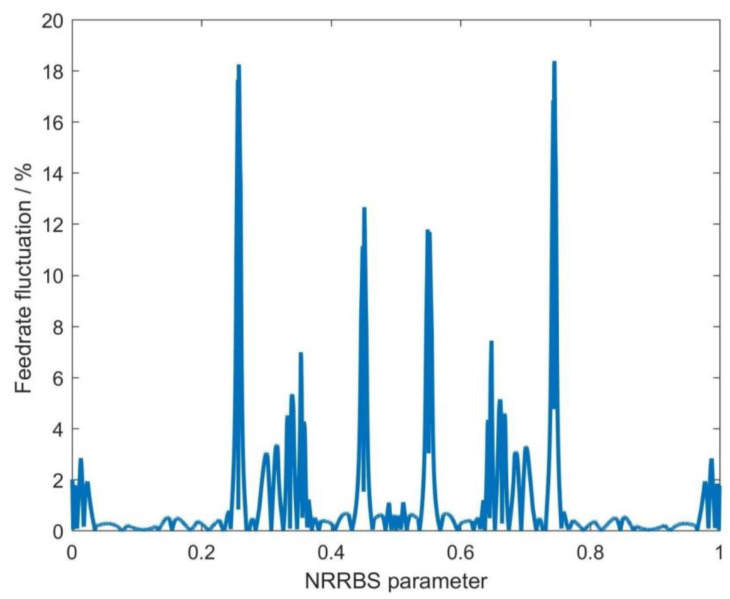
The feedrate fluctuation of the first-order adaptive feedrate method (FAM).

**Figure 6 sensors-23-03789-f006:**
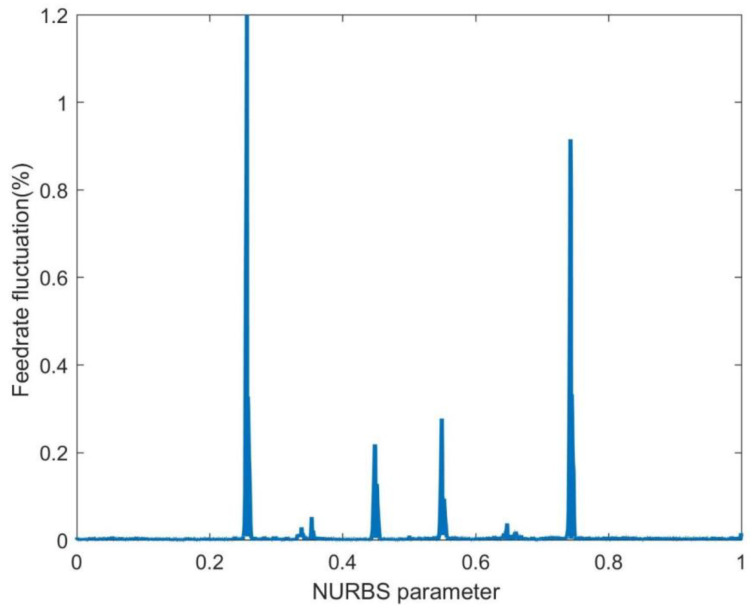
The feedrate fluctuation of the FAM with the first-level parameter compensation.

**Figure 7 sensors-23-03789-f007:**
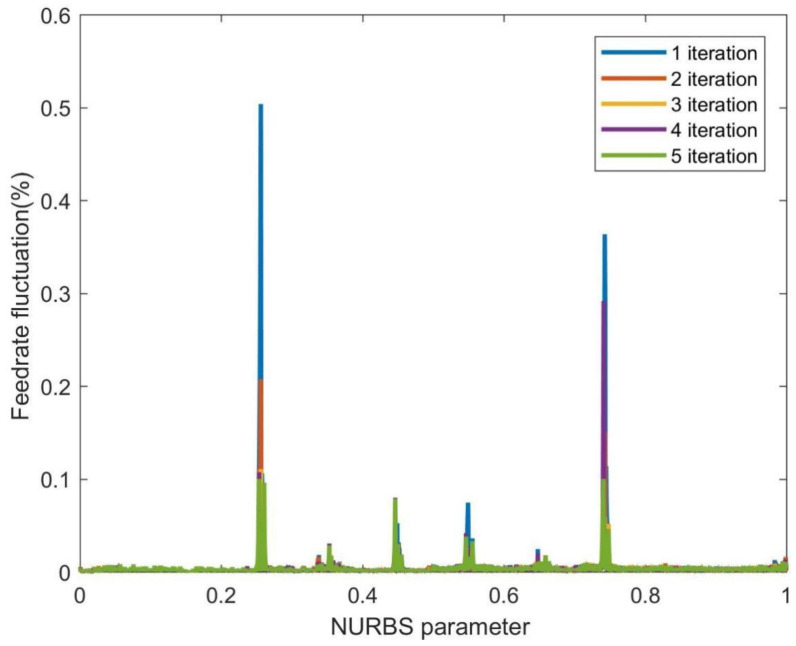
The feedrate fluctuation of the FAM with the two-level parameter compensation.

**Figure 8 sensors-23-03789-f008:**
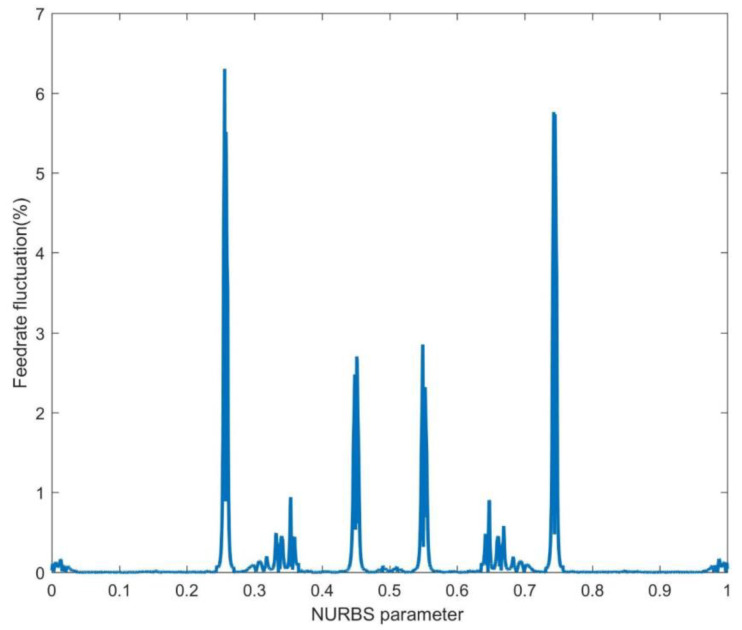
The feedrate fluctuation of the second-order adaptive feedrate method (SAM).

**Figure 9 sensors-23-03789-f009:**
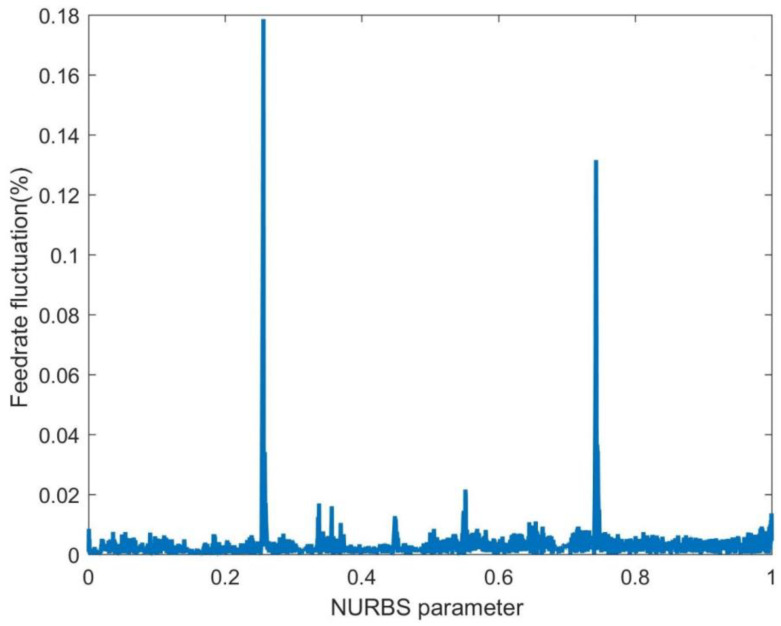
The feedrate fluctuation of the SAM with the first-level parameter compensation.

**Figure 10 sensors-23-03789-f010:**
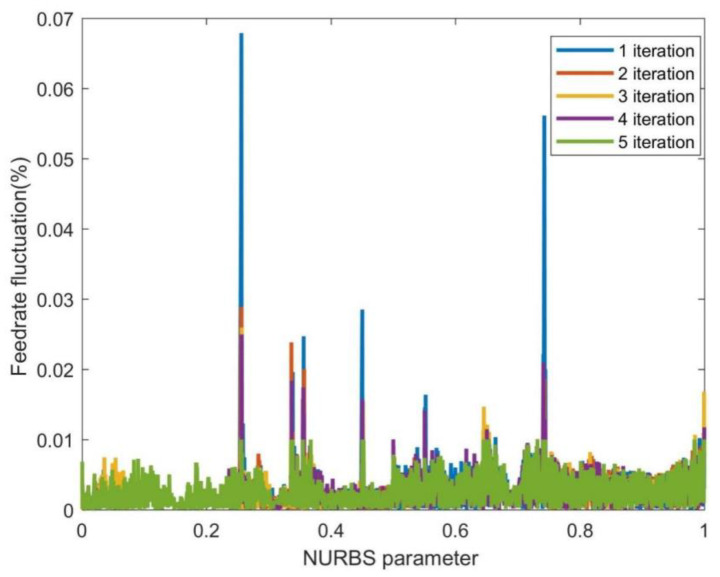
The feedrate fluctuation of the SAM with the two-level parameter compensation.

**Table 1 sensors-23-03789-t001:** Simulation results of the proposed parameter compensation interpolation algorithms.

IA	MFFR (%)	SFF (Σ*ε_i_*,%)	ACT (us)
FAM	18.39	2.2484×10³	135
FAM with FLPC	1.20 (↓93.5%)	18.14 (↓99.19%)	290
FAM with TLPC (iteration 1)	0.50 (↓97.3%)	9.86 (↓99.56%)	302
FAM with TLPC (iteration 5)	0.10 (↓99.5%)	6.59 (↓99.71%)	350
SAM	6.32	269.41	155
SAM with FLPC	0.18 (↓97.2%)	6.44 (↓97.10%)	310
SAM with TPPC (iterations 1)	0.07 (↓98.9%)	5.71 (↓97.88%)	322
SAM with TLPC (iterations 5)	0.01 (↓99.8%)	5.40 (↓98.00%)	360

IA—interpolation algorithms. MFFR—the maximum feedrate fluctuation rate. SFF—the sum of feedrate fluctuation in the whole interpolation process. ACT—average computational time.

**Table 2 sensors-23-03789-t002:** Simulation results of different interpolation algorithms.

Interpolation Algorithms	The Maximum Feedrate Fluctuation Rate (%)
Proposed SAM with TLPC	0.01
IRF method [26]	2 (↑1.99 × 10^4^%)
Fast NURBS (fifth-degree ILF) [29]	0.025 (↑1.5 × 10^2^%)
TFSS [12]	0.25 (↑2.4 × 10^3^%)
OFB [39]	0.027 (↑1.7 × 10^2^%)

## Data Availability

The data presented in this study are available on request from the corresponding author.

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
