# Peer review of "NURBS Interpolator with Minimum Feedrate Fluctuation Based on Two-Level Parameter Compensation"

_sensors, 2023, doi:10.3390/s23083789_

Round 1

Reviewer 1 Report

In this paper, the authors shows Minimizing the feedrate fluctuation is the main task of an accurate NURBS interpolator,  significantly affecting the machining quality. Considering the sensitivity of the curvature-sensitive  area to feedrate fluctuations, this paper proposes a two-level parameter compensation method for  the NURBS interpolator to minimize the feedrate fluctuation. First, to overcome the federate fluctuation in the non-curvature sensitive areas with low calculation cost, the first-level parameter compensation is performed, and the parameter compensation is realized by the Taylor series expansion method so that the chord trajectory of the new interpolation point after compensation is as close as possible to the original arc trajectory. The idea behind this is interesting. However, I still have quite a number of concerns in this manuscript. There are times where there are not enough data to support the conclusions of the author. Please see some of the major concerns below.

1.The information for the flowchart of the parameter compensation is not enough. The authors should give much more information about this for example can we use a deep learning algorithm for solving this parameter?

2.  The authors should give much more information about the novelty of this paper, especially the effect of minimizing the feedrate fluctuation, which applications can be used this method and how for example how you can integrated this process in super-resolution application ?

3. More references need to be included in the introduction part to understand the applications of decoding in interpolation method in super-resolution method for example.

a.     Improving Raman spectra of pure silicon using super-resolved method

- Journal of Optics, 2019

b.     Super-resolved Raman spectroscopy

- Spectroscopy Letters, 2013

5.  Much more discussion about the results should be given in this paper, especially the author needs to provide enough physicals mechanism analysis about the results.

Reviewer 2 Report

(1) The abstract of this manuscript needs to be improved. The background description of the problem proposed is a bit too brief, and the value of the problem proposed in the field of engineering technology should be highlighted.

(2) The two-level parameter compensation method proposed in this manuscript is impressive, innovative, even if it is relatively simple.

(3) Richer simulation results or examples should be given in the revised edition.

Reviewer 3 Report

1) It is suggested to improve the research motivation in the first paragraph of the Introduction. For example, what is the demanding application of the NURBS interpolator? Clearly define this part will assist the reader to appreciate the work.

2) Table 2 should be in one page.

3) Figure 8 to 10 are located before the text cited it.

Reviewer 4 Report

Dear author

Based on my evaluation, the topic of the paper is extremely interesting and relevant, however some minor revisions are required in order to improve this work.

- The abstract could include the core gap in the body of knowledge [literature review] and this will strengthen the purpose section and rationale for the study.

- The research objectives could be included at the end of the literature review/context section and before the methods section and this will remind the reader of the objectives but also act as clear signposting to the methods section.

- I recognize that the paper was guided by some theoretical fundamentals relevant for this subject.

In fact, the author was very accurate in analysing the selected topic and other related concepts.

- The introduction section is very useful to understand the whole work; it establishes the importance of research, establishes a theory-based gap, research question, some contribution, and the paper’s structure.

- The author presents original ideas on the topic and aims at demonstrating critical thinking and ability to draw conclusions based on the knowledge of relevant theory and relevant empirical material.

- I liked the fact that the positioning of the study is presented in the introductory section, identifying important gaps in the literature and considering some contributions to existing theoretical knowledge.

- Definition for the constructs are well structured.

- Moreover, there is a distinct value added of the paper.

Thus, the author explains why the observed phenomena occurred and why it could be necessary to analyse the phenomenon at this stage.

- The paper contributes to existing research, and it is interesting and relatively new of its kind, however the limitations and future research should be enriched!

- Overall, the paper is pretty much consistent with a good range of literature/sources from classic to recent publications. Plus, the methods section was quite detailed in terms of piloting and analysis which hopefully provides rigor to the study.

Best Regards

Round 2

Reviewer 1 Report

The new version can be published.